# Physiology, Diagnosis and Treatment of Cardiac Light Chain Amyloidosis

**DOI:** 10.3390/jcm11040911

**Published:** 2022-02-09

**Authors:** Agnieszka Stelmach-Gołdyś, Monika Zaborek-Łyczba, Jakub Łyczba, Bartosz Garus, Marcin Pasiarski, Paulina Mertowska, Paulina Małkowska, Rafał Hrynkiewicz, Paulina Niedźwiedzka-Rystwej, Ewelina Grywalska

**Affiliations:** 1Department of Immunology, Faculty of Health Sciences, Jan Kochanowski University, 25-317 Kielce, Poland; astelmach3@o2.pl (A.S.-G.); bartosz.garu@gmail.com (B.G.); marcinpasiarski@gmail.com (M.P.); 2Department of Hematology, Holy Cross Cancer Centre, 25-734 Kielce, Poland; 3Department of Experimental Immunology, Medical University of Lublin, 20-093 Lublin, Poland; monika.zaborekk@gmail.com (M.Z.-Ł.); jakublyczba7@gmail.com (J.Ł.); paulinamertowska@gmail.com (P.M.); ewelina.grywalska@umlub.pl (E.G.); 4Institute of Biology, University of Szczecin, 71-412 Szczecin, Poland; paulina.malkowska@phd.usz.edu.pl (P.M.); rafal.hrynkiewicz@usz.edu.pl (R.H.)

**Keywords:** amyloidosis, cardiac amyloidosis, light-chain amyloidosis, molecular mechanisms, protein aggregation, misfolding

## Abstract

AL (light-chain) amyloidosis is a systemic disease in which amyloid fibers are formed from kappa or lambda immunoglobulin light chains, or fragments thereof, produced by a neoplastic clone of plasmocytes. The produced protein is deposited in tissues and organs in the form of extracellular deposits, which leads to impairment of their functions and, consequently, to death. Despite the development of research on pathogenesis and therapy, the mortality rate of patients with late diagnosed amyloidosis is 30%. The diagnosis is delayed due to the complex clinical picture and the slow progression of the disease. This is the type of amyloidosis that most often contributes to cardiac lesions and causes cardiac amyloidosis (CA). Early diagnosis and correct identification of the type of amyloid plays a crucial role in the planning and effectiveness of therapy. In addition to standard histological studies based on Congo red staining, diagnostics are enriched by tests to determine the degree of cardiac involvement. In this paper, we discuss current diagnostic methods used in cardiac light chain amyloidosis and the latest therapies that contribute to an improved patient prognosis.

## 1. Introduction

Systemic amyloidosis is a disease caused by the deposition of abnormally folded fibrous proteins in extracellular spaces in various tissues and organs [1]. To date, 37 precursor proteins have been identified that can undergo molecular transformation and form amyloid fibrils in humans [2]. The main cause of mortality in this disease is cardiac involvement. Two types of amyloidosis are known to infiltrate this organ: light chain amyloidosis (AL), transthyretin amyloidosis (ATTR), and sometimes, in acquired amyloidosis, type A (AA) cardiac involvement may occur [3]. Depending on the type of amyloidosis, the clinical phenotype can vary considerably. AL amyloidosis is the most commonly diagnosed and occurs with a frequency of about 6–10 cases per million. AL, formerly called primary amyloidosis, is a clonal disorder of plasma cells caused by overproduction and abnormal folding of antibody light chain fragments [4]. Cardiac involvement occurs through extracellular amyloid infiltration in the myocardium, which causes thickening of the walls of both chambers. This contributes to excessive fluid accumulation in the body known as congestive heart failure [5]. This review paper will discuss the pathophysiology of cardiac light chain amyloidosis and the current methods used to diagnose and treat this condition.

## 2. Characteristics of Amyloidosis

Amyloidoses are a family of diseases that cause abnormal folding of precursor proteins that assemble into amyloid fibrils [6]. The lesions are caused by the deposition of these fibrils, forming amyloid plaques in systems and organs [7]. To date, 37 proteins and peptides have been identified that are capable of forming amyloid deposits in humans in vivo [2], but it can also be produced in vitro [8]. Depending on the type of precursor protein, there is a different type of systemic amyloidosis (Table 1).

It is important to note that several proteins can form a functional amyloid and that not all amyloids are pathological [10]. A common feature of all systemic amyloidoses is that the precursor protein is expressed in one or more tissues, transported through the bloodstream, and eventually deposited in target organs to form amyloid fibrils [11]. The deposition of amyloid fibrils causes cellular stress and changes in tissue architecture, which can result in organ dysfunction and even death [12]. Amyloid formation can be triggered not only by an increase in precursor protein concentration in body fluids but also by mutations that promote abnormal folding. It has been suggested that Alzheimer’s disease may be caused by excessive production of amyloid precursor protein and insufficient removal [11]. Amyloid deposits appear, mainly, in the elderly. This phenomenon has not been fully explained, but it is suspected to be due to impaired repair mechanisms and failures of the intra- and extracellular proteostatic apparatus [7,12]. The average age at diagnosis is 63 years, but in 1.3%, the diagnosis is made below 34 years of age [13,14]. Males account for 55% of all amyloidosis patients.

Amyloid fibers comprise the precursor amyloid protein, which is about 90%, as well as other extracellular matrix proteins (10%) [15]. The participation of these proteins in the formation of amyloid fibers consists of stabilization and prevention of degradation [16].

Among them, we can distinguish: serum amyloid P (SAP), apolipoprotein E (ApoE), apolipoprotein A1 (ApoA1), and apolipoprotein A4 (ApoA4). SAP is a amyloid protein (a component of serum P amyloid), which is found in all types of amyloid, has a molecular weight of 254 Da, is both synthesized and catabolized in the liver. SAP acts as a binder for amyloid fibers, which may additionally protect amyloid from proteolytic factors [17,18]. The mechanisms of amyloid formation are composed and still not fully understood. It is considered that the amyloidogenesis process is influenced by both the type of light chain, reactions with extracellular proteins, and the direct effect on cells. Amyloid fibrils can be observed by electron microscopy (EM) or atomic force microscopy (AFM), where they are visible as thread-like structures approximately 7–13 nm in diameter. Their structure consists of 2–8 protofilaments that twist around each other and connect laterally to form flat ribbons [7]. The filaments have β-bands that run perpendicular to their axis and are connected in β-sheets [19]. This structure can be observed using staining techniques such as thioflavin-T (ThT) fluorescence, Congo red (CR) binding, or their derivatives. These dyes form ordered arrangements along the fibers, giving a specific spectral response. Another popular technique for amyloid identification is Fourier transform infrared (FTIR) spectroscopy [8]. Comparison of the material properties of amyloid fibrils with structures of biological origin and synthetic polymers has shown that amyloid fibrils are stiffer than most biological intracellular structures [6].

## 3. Light-Chain (AL) Amyloidosis

In the context of amyloidosis, the most common systemic disease is light chain (AL) amyloidosis, which is caused by plasma cell clones. Amyloid fibrils deposited in tissues are formed by the antibody light chain (LC), which is a precursor protein [20]. Immunoglobulins secreted by plasma cells consist of two light chains and two heavy chains that are linked by covalent bonds through disulfide bridges [21]. The light chains of immunoglobulins, and especially their variable parts, are involved in the formation of fibril proteins in primary amyloidosis or amyloidosis in the process of multiple myeloma. Some light chain subtypes are more prone to fibril formation. Such observations were carried out as early as the second half of the 20th century [22], asserting the presence of monoclonal lambda chains, more often than kappa, in amyloid deposits in patients with primary amyloidosis, whereas monoclonal kappa chains were found more often in patients with multiple myeloma or B-cell lymphoproliferation who were not diagnosed with amyloidosis [22]. The distinctive characteristics and types of light chains associated with them depend on the structure of their variable parts. The subtypes of both λ and κ chains are determined from the amino acid sequence of the first framework region (FR1) of the N-terminal segment of the variable part of an immunoglobulin light chain (VL). Among the lambda chains, based on the analysis of the structure of proteins isolated from monoclonal immunoglobulins, six subtypes were distinguished and classified as VλI, VλII, VλIII, VλIV, VλV i VλVI. Research conducted by Solomon et al. [23] and published in 1982 allowed for identifying the structure of VλVI by isolating a protein with a filamentous structure from the spleens of two patients with primary amyloidosis and comparing it to the constitution of the variable lambda chains in patients with the presence of monoclonal protein, however, without amyloidosis.

The amyloidogenic clone is often characterized by chromosomal abnormalities. The most common cytogenetic abnormalities found in AL amyloidosis are: t (11; 14) (q13; q23), which occurs in 50% of cases, monosomy 13/deletion (13q14) (36%), and trisomies, which occur in 26% [17,24,25,26].

AL amyloidosis is a rare disease associated with the dyscrasia of plasma cells that produce monoclonal free light chains that form β-sheet amyloid fibrils that can accumulate extracellularly in tissues and organs, resulting in impairment of their function [27]. It is diagnosed with a frequency of approximately 1 in 100,000 per year [28]. The diagnosis and treatment of AL presents a challenge to medicine. This is often due to delayed diagnosis, which is time consuming because it requires a complete clinical picture to determine the type of amyloidosis we are dealing with [9]. This is because the symptoms are non-specific and may vary from patient to patient. There is no single diagnostic test to make a diagnosis, so the key person during diagnosis is a physician who must have a high index of suspicion and will order further tests. In most cases, diagnosis is so late that organ damage is too advanced and impairs the ability to treat effectively. Organ dysfunction in AL amyloidosis is caused not only by the deposition of amyloid fibrils but also by the direct action of cytotoxic amyloidogenic light chains [29].

### 3.1. Molecular Mechanisms

As we mentioned earlier, AL ameloidosis is caused by a small clone of B cells that produce a light chain with mutations in the variable region. These mutations promote improper protein aggregation and cause folding [11]. Organ dysfunction can result from amyloid deposits disrupting tissue structures or from proteotoxicity [30,31]. Cellular toxicity is determined by oligomers (Figure 1) that are formed through interactions with the extracellular environment, e.g., chaperone proteins, matrix components, endoproteases, and metals. Both misfolded proteins and oligomers can induce cellular stress, disrupt tissue function and exacerbate pathological changes [10,11]. In X-rays, fibers can be observed forming a cross diffraction pattern of β-fibers formed through oligomers. Their formation may be catalyzed by amyloid fibrils [11]. Marin-Argany and co-authors [32] report that amyloid fibrils, at low concentrations, exhibit cytotoxicity, and amyloid LCs contribute to apoptosis. Further directed studies in AL amyloidosis proved an association between genes encoding light chain variable parts and organs affected by amyloid deposits. The germline gene LV6-57 was observed to be more prevalent in AL amyloidosis and to be associated with renal involvement. The LV1-44 gene is associated with cardiac involvement, and KV1-33 is associated with liver involvement [33,34].

### 3.2. Symptoms of AL Amyloidosis

The clinical symptoms of AL amyloidosis depend on the damage to the affected organs but are usually not specific. Amyloidosis is defined as a disease in which the patient’s general condition deteriorates for no apparent reason. The most common symptoms are weight loss, diarrhea, fatigue, and exercise dyspnea [13,35]. Due to the initially slow course and symptoms resembling other disease entities, the diagnosis is usually made quite late, with advanced organ changes most often affecting the heart (71%), kidneys (58%), gastrointestinal tract (22%), nervous system (23%), liver (16%), and soft tissues (10%) [13,36].

While it is known that amyloid deposits can affect both small and large arteries of the heart, kidneys, and the gastrointestinal tract and lead to dysfunction, the cellular mechanisms leading to amyloid angiopathy are not fully understood [37]. The symptoms associated with amyloid angiopathy are diverse and may take place under the forms of ischemic disease and myocardial infarction, arrhythmias, malabsorption, proteinuria, and renal failure [38].

The model of the formation of amyloid deposits assumes the participation of medial smooth muscle cells in which beta-protein precursors accumulate. They are then the source of the β-amyloid found in the early stages of amyloidosis in the extracellular deposits of blood vessels. This suggests the role of smooth muscle cells in the vascular median membrane in the formation of amyloid, the presence of which is observed in the vascular walls in the systemic circulation [37,39]. It is worth noting that amyloid deposits do not occur in small capillaries, which suggests the participation of smooth muscle cells vasculature in the formation of amyloid [37].

At the time of diagnosis, the kidneys are involved in 50–80% of patients with AL amyloidosis. In the Polish population over 65 years of age, the frequency of AL amyloidosis diagnosis is 2.8%, and in people aged 18–64 years, AL amyloidosis was diagnosed in 1.1% of cases [40]. This percentage is comparable to the literature, where the administered amount is 0.21–1.00% [41]. The symptoms of amyloidosis include various degrees of proteinuria (30% of patients develop nephrotic syndrome) and impaired renal function (renal failure affects 50% of respondents) [42].

The clinical picture depends on the location of amyloid deposits and in cases where the glomeruli are involved, proteinuria dominates, while when the vessels and interstitial tissue of the kidneys are affected, renal function is impaired. In the case of AL amyloidosis, nephrotic syndrome is more common, which indicates greater damage to the basement membrane. Diagnostic tests should include diurnal proteinuria and urine immunofixation [43]. Dyslipidemia and hypoalbuminemia can be detected in other additional tests, and in enlarged kidneys on imaging tests. The risk of progression to end-stage renal disease depends on the amount of proteinuria at diagnosis and the amount of estimated glomerular filtration rate (eGFR) [44].

Deposits of amyloid in soft tissues may result in macroglossia, hoarseness, obstructive night apnea, spectacle hematomas after minor trauma or exertion, swelling of the submandibular salivary glands, xerostomy, or periarticular infiltration of soft tissues or synovium. Carpal tunnel syndrome may precede the diagnosis of AL amyloidosis by several years (from 1 month to over 9 years) [13].

Mixed sensory-motor neuropathy manifests itself in approximately 20% of patients, and disturbance of the autonomic nervous system is observed in 15% of patients with amyloidosis. Symptoms such as numbness, paraesthesia, and pain are the result of involvement of peripheral nerves. On the other hand, dysfunction of the bladder and intestines, as well as orthostatic hypotension, are the result of damage to the autonomic nervous system [13].

Coagulation disorders may occur in AL as hemorrhagic diathesis and plasma coagulation disorders. In a study conducted by Mumford et al. [45], 337 patients with AL were involved. In 28% of them, bleeding disorders were found, while as many as 51% of patients presented abnormalities of the plasma coagulation system. Abnormal activity of factor X was also confirmed in 22 out of 154 examined patients (14%). The most common disorder observed was an increase in thrombin time (32%), followed by an increase in prothrombin time (24%). Prolongation of prothrombin time coexisted with amyloid infiltration of the liver, proteinuria, and decreased serum albumin, while thrombin time extension was independent of the total amount of amyloid determined by SAP scintigraphy. Additionally, it is suggested that amyloid infiltration in the walls of blood vessels has an increased bleeding tendency in patients with AL.

### 3.3. Cardiac AL Amyloidosis

As previously mentioned, cardiac light chain amyloidosis (AL-CA) leads to the deposition of amyloid fibrils in the heart, causing thickening of both chambers [46]. The pattern of amyloid deposition is usually subendocardial and diffuse. They are localized in the interstitium, surrounding the myocytes, or in the intramural coronary arteries. Amyloid is present throughout the myocardial tissue [47]. Usually, the valves are thickened causing regurgitation. During AL-CA, there may be small pericardial effusions and coronary involvement, causing ischemia and angina pectoris, with normal coronary vessels [48]. Amyloid deposition on the ventricular walls makes them stiff and contributes to diastolic filling abnormalities. During severe or high-grade disease, marked systolic dysfunction is evident [4].

Cardiac involvement is the major negative prognostic factor, as 75% of deaths are due to heart failure or arrhythmias associated with AL amyloidosis [49]. The prognosis without treatment is severe, with a survival of less than one year [50]. Amyloidosis affects the heart in more than 90% of systemic cases, but it should be remembered that, in about 5% of patients, the heart is the only organ involved [51]. Amyloid fibers can infiltrate all structures of the heart muscle, leading not only to thickening of the ventricles, but also to relaxation disorders and, consequently, restrictive cardiomyopathy [49]. The conductive system is also occupied by amyloid, which may result in arrhythmias [52].

Atrial fibrillation occurs in 10–15% of patients with cardiac amyloidosis [51]. Blood pressure is lowered, and often in people with previously diagnosed hypertension, there is “spontaneous recovery” and a drop in blood pressure. Another symptom that occurs is orthostatic hypotonia, especially in the presence of amyloid of the autonomic nervous system and recurrent pleural effusion, caused by both cardiovascular failure in the course of amyloidosis and pleural infiltration by amyloid [53]. In electrocardiographic examination, (ECG) low voltage of QRS [28] complexes can be found in up to two-thirds of patients, but the echocardiographic examination provides a lot of information already in the initial stage of the disease. It can also be used to assess prognosis by measuring the deceleration time of the rapid wave of left ventricular filling [54].

## 4. Diagnostics

In order to improve outcomes for patients with the deadly disease cardiac amyloidosis, it is critical to educate physicians and familiarize them with the latest diagnostic methods. Early diagnosis of amyloidosis offers the best chance of improving the prognosis of patients, especially when the disease has not progressed enough to occupy the heart [55]. Studies have shown that patient survival, when the heart is already affected, is very low [56]. In addition, the emergence of newer treatments for cardiac amyloidosis also contributes significantly to patient survival [55].

Diagnostic criteria for AL amyloidosis according to IMWG (International Myeloma Working Group) guidelines:Presence of clinical syndromes related to the affected organsPositive result Congo red staining of tissue specimensIdentification of light chains in amyloid depositsPresence of deviations in additional studies resulting from the proliferation of monoclonal plasma cells

Amyloidosis, as the cause of the presented clinical picture, should be included in the differential diagnosis in the presence of nephrotic syndrome in a patient without a diabetes diagnosis, cardiomyopathy that is not a result of ischemic heart disease, liver enlargement, with preserved parenchyma structure or conditions, in which monoclonal protein in the serum and neuropathy coexist or we observe laboratory features of monoclonal gammapathy with undetermined significance, accompanied by weight loss and a feeling of fatigue. The basis of the diagnosis of amyloidosis is the presence of amyloid in the tissue suspected of the presence of organ deposits or alternative sites where amyloid accumulation also occurs in the case of systemic amyloidosis (abdominal fat or mucosa of the rectum, gums, and salivary glands) or histopathological examination of the bone marrow [13,57]. Subcutaneous fat aspiration biopsy, which is a minimally invasive and safe procedure, allows one to determine the presence of amyloid deposits with 80–94% sensitivity. In the case of biopsy of the remaining tissues mentioned, the sensitivity is 75–85%, respectively [58]. Performing a simultaneous adipose tissue biopsy and bone marrow biopsy with histopathological examination allows for diagnosis in 85% of patients. If it is not possible to make a diagnosis on this basis, it is necessary to perform a biopsy of the affected organ. Congo red staining of the examined tissue is a diagnostic standard and a necessary condition for diagnosis. Amyloid is visible in the light microscope as red-orange fibers, and it is visible in the polarized microscope as light green birefringent deposits. It should be remembered that amyloid accumulates in the tissue focally and, therefore, it is necessary to evaluate many preparations [59]. The next diagnostic stage is the differentiation of the amyloid type, for which immunohistochemical and immunofluorescence methods are used, with the use of antibodies that bind kappa and lambda chains. Mass spectrometry is used to precisely determine the type of proteins that make up amyloid. Indications for the use of this diagnostic method in patients with AL amyloidosis include no tissues available for immunofluorescence (IF) testing, negative, or double-positive reactions towards both kappa and lambda chains, strong positive IF towards heavy immunoglobulin chains, with or without staining light chain, and ambiguous Congo red staining [43].

Disease activity and response to treatment can be assessed by measuring monoclonal protein levels in serum and urine. Due to its low diagnostic sensitivity and low concentration of monoclonal protein, electrophoresis is insufficient in about half of the patients, when the “peak” of the monoclonal protein is not detected [43]. On the other hand, in the presence of a monoclonal protein, a typical pattern of electrophoresis is characterized by a decrease in the albumin and γ-globulin fractions and an increase in the α2-globulin fraction. All patients should also be immunofixed with serum and urine proteins. This method enables the determination of a monoclonal protein present at a concentration of approximately 150 mg/L [60]. The test that is characterized by the highest sensitivity in the diagnosis of AL amyloidosis is the determination of free light chains (FLC) in serum (88%), and additional urine immunofixation increases the sensitivity to 98%. It is a quantification using nephelometry, which allows measurement at FLC concentrations of ≈ 10 mg/L [61]. This test is also used to assess the possible progression of the disease and monitor the response to treatment. In addition to the concentration of free light chains κ and λ, the ratio of their concentrations κ/λ is important, which is correctly 0.26–1.65. It is an index of the clonality produced by pathological plasmocytes of one of the chains. In the assessment of prognostic factors, an indicator called dFLC is taken into account. This is the difference. The next step in the diagnosis is to perform a bone marrow biopsy in order to exclude other plasma cell dyscrasias. It should be remembered that they may occur with, or precede, the development of AL amyloidosis.

In the case of histopathological examination of bone marrow, prognostic significance was found depending on the percentage of plasma cells. The percentage above 10% deteriorates significantly, similarly to the diagnosis of multiple myeloma [62].

The relationship between the light chain type and the organs affected by amyloid deposits was confirmed by the study of Kumar et al. [63]. Among 730 patients diagnosed with AL amyloidosis, 72% had a monoclonal λ chain, and the remaining 28% had κ. In the first group, renal involvement was predominant, while changes in the gastrointestinal tract and liver were observed in κ-AL patients. There was no difference in overall survival between λ-AL and κ-AL. However, it was shorter in patients with a higher value of dFLC. According to observational studies conducted by Dispenzieri et al. [64], among 119 patients with AL amyloidosis who underwent bone marrow stem cell transplantation, the concentration of monoclonal light chain may be a prognostic indicator of survival. It can also be used to assess both the hematological response and the improvement of the functions of the affected organs. It should be remembered that, in the case of renal failure, the concentration of polyclonal light chains increases, and then, the FLC ratio becomes more prognostic. However, in the case of significant immunosuppression of the polyclonal chain, the FLC ratio value is not prognostic.

### 4.1. Diagnosis of Cardiac Amyloidosis

Although cases of cardiac involvement in AL amyloidosis are common, AL-CA was considered a rare disease with respect to the general population. However, recent data on AL-CA suggest that it is underestimated and may cause common heart disease [65]. For systemic amyloidosis, cardiac involvement is the most important organ-related prognostic factor. Therefore, evaluation of cardiac function is the cornerstone of diagnosis and is most important [29].

A very useful biomarker to assess myocardial dysfunction arising from AL amyloidosis is NT-proBNP. It is the most sensitive indicator of cardiac toxicity caused by amyloidogenic light chains [66,67]. Brain natriuretic peptide (BNP) is used not only for diagnostic evaluation but also to assess the cardiac response to treatment. Circulating BNP levels can be reduced in parallel with a decrease in free LC (FLC) after chemotherapy, and such reductions translate into improved survival [68]. The same measurements and troponin can also be used to determine the extent of myocardial damage [69].

Cardiac imaging is another important component during the diagnosis of AL-CA. Echocardiography provides a sensitive tool to evaluate the heart. The classic image indicating myocardial involvement is myocardial thickening and a speckled pattern. The heart is considered affected when the mean left ventricular wall thickness is greater than 12 mm (in the absence of other explanations) [29]. However, it has been observed that the walls of the heart may remain normal, despite its dysfunction, in 3–30% of cases [70,71]. Doppler echocardiography is used in the early detection of heart failure in AL amyloidosis. Symptoms of heart failure usually develop with preserved ejection fraction, so the dysfunction is mainly diastolic. Doppler echocardiography assesses the ability of the heart to shorten when contracting in the longitudinal plane. As reported in the literature, a negative predictor of patient survival is a negative longitudinal systolic basal strain of the basal anteroseptal segment [72].

An electrocardiogram (ECG) may be another diagnostic test for AL-CA. While cardiac involvement is present, the ECG often shows low voltage with an abnormal axis [56]. The P-wave usually has normal voltage, but it often has abnormal morphology, and it is often significantly prolonged despite low voltage QRS complexes. This ECG finding represents slowed atrial conduction due to amyloid infiltration. A low-voltage ECG often occurs before heart failure develops and before the left ventricular wall thickness increases on echocardiogram, so it may be an early marker of disease [73]. The discrepancy between increased left ventricular mass on the echocardiogram and low voltage on the ECG should raise the suspicion of cardiac amyloidosis [56].

Magnetic resonance imaging (MRI) is another diagnostic test for detecting AL-CA. It has the advantage of higher resolution and the ability to further characterize the myocardium using late gadolinium enhancement (LGE) imaging and T1 mapping, which is a quantitative technique capable of detecting diffuse myocardial abnormalities, including amyloid burden. It is an excellent diagnostic tool characterized by late LGE enhancement, reflecting interstitial edema due to the presence of amyloid deposits [29]. Post-contrast T1 mapping decreases when myocardial tissue reaches zero transition at an earlier or similar inversion time as the blood pool, resulting in an increased extracellular volume (ECV) generally >40%. Neither MRI nor echocardiography can differentiate between types of amyloidosis [55]. Furthermore, because renal impairment is common in AL amyloidosis, injection of gadolinium may be harmful. It can cause debilitating renal systemic fibrosis in patients, so it is recommended to perform echocardiography with stress imaging in all patients, which also has strong prognostic significance, and perform MRI in cases where cardiac involvement is still questionable [29].

### 4.2. Red Flags

Cardiac amyloidosis produces many symptoms, including extracardiac symptoms. A large number of these are extremely useful for suspecting the disease and are called “red flags.” These include many symptoms such as bruising of the skin, proteinuria, deafness, bilateral carpal tunnel syndrome, ruptured biceps tendon and many others. Unfortunately, not all of these are related to AL amyloidosis, but along with cardiac symptoms, they may provide an opportunity for earlier detection of the disease (Table 2) [55,65].

## 5. Prognosis

Currently, AL amyloidosis has the worst prognosis of all types of systemic amyloidosis. If treatment is not instituted for some reason, the median survival is 8 months. This can be caused by both light chain toxicity and organ involvement by amyloid fibrils. Prognosis may vary depending on the severity of the disease and the burden of the deposit with particular attention to cardiac status [29].

Cardiac involvement is a major predictor of survival time (Table 3). The severity of the lesions, determined according to the Mayo Clinic classification, is assigned 1 point for subsequent deviations in additional tests: dFLC ≥ 18 mg/dL, cTnT ≥ 0.025 ng/mL, NT-proBNP ≥ 1800 pg/mL. This corresponds to stages I–IV, with 0–3 deviations, respectively. In each group, the overall survival is 94.1; 40.3; 14.0, and 5.8 months [74]. A conversion tool is available to use the above models with alternative cardiac biomarker assays (BNP, troponin I, and high-sensitivity troponin T) [9].

## 6. Treatment

Treatment of cardiac amyloidosis varies depending on the type of amyloid and response to treatment. They can be divided into three types: (I) supportive therapy, (II) therapy to inhibit the production of the corresponding precursor protein, and (III) novel strategies to inhibit amyloid fibril formation or to directly target amyloid deposits or stabilize the precursor protein. The third type of treatment is used especially in ATTR. With some carefully selected patients, heart transplantation is possible, although it is rarely performed. It should be emphasized, however, that despite the many options (Table 4), the prognosis of patients diagnosed with cardiac amyloidosis is low [3].

### 6.1. Supportive Therapy

Supportive treatment is based on modified heart failure treatment (including devices). These modifications are necessary because standard treatment can make the AL-CA patient worse. There is little information regarding the use of conventional medications (beta-blockers, angiotensin-converting enzyme inhibitors, and angiotensin receptor blockers), but they are claimed to have caused a decrease in cardiac minute volume in many cases, which exacerbates symptoms [3]. Patient education, regarding diet and medication intake, is important in treatment. Maintaining adequate blood pressure, balancing peripheral edema, and renal insufficiency are key. To do this, a balance between salt restriction and water supply and diuretics is essential [1]. The use of pacemakers and implantable cardioverter-defibrillators are also ineffective and do not prevent sudden cardiac death, as this is often caused by electromechanical dissociation [75]. Orthostatic hypotension can be treated with midodrine and compression stockings, whereas fludrocortisone only enhances fluid retention [1].

### 6.2. Therapy to Inhibit the Production of the Corresponding Precursor Protein

The mainstay of AL-CA treatment is chemotherapy, which targets a core population of clonal plasma cells. The most common treatment is dexamethasone in combination with an alkylator. The efficacy of cyclophosphamide and dexamethasone is improved by the addition of thalidomide; however, this likely increases toxicity [76]. At present, dexamethasone appears to be the most useful agent, but must be used with caution as it increases the risk of fluid overload and requires a change in diuretic therapy [77]. Another promising agent is bortezomib, combinations of which appear to be effective in amyloidosis [78,79,80]. However, with any treatment, it is important to select a hematologist who specializes in AL amyloidosis and a cardiologist who will select the appropriate agent and dosage to minimize side effects [5]. In specially selected patients, high-dose chemotherapy with autologous stem cell transplantation (ASCT) is also an option. This therapy is used in patients in whom oral regimens have failed. Eligibility is based on the patient’s overall fitness and organ function, with primary emphasis on cardiac function [56]. Patients must meet Mayo eligibility criteria for ASCT which include: physiologic age ≤ 70; performance score ≤ 2; NYHA class I/II; cardiac troponin T < 0.06 ng/mL; systolic blood pressure > 90 mm HG; creatinine clearance ≥ 30 mL/min/1.73 m^2^; less than three organs significantly involved [29]. Outcome studies in transplant patients showed a current five-year survival rate of 77% [81].

### 6.3. Latest Treatment for AL Amyloidosis

Due to the fact that ASCT is used only in some patients, and other available methods do not give as satisfactory results as we would like, the search began for other means that would effectively improve patient outcomes. Preliminary clinical trials (Table 4) have shown that the use of daratumumab in the treatment of AL amyloidosis can contribute to complete hematologic remission and normalization of light chains [82]. Daratumumab is a human IgG1k monoclonal antibody (mAb) that has a high affinity for the CD38 epitope found strongly on the surface of multiple myeloma cells [83]. Daratumumab induces myeloma cell death contributing to reduced mortality among affected patients. Due to the fact that abnormal cell clones in AL amyloidosis also have CD38 on their surface, the possibility of using daratumumab in the treatment of this disease has been considered [82]. The first report of its use was in two patients diagnosed with AL amyloidosis. Previous treatment attempts with chemotherapy and ASCT showed resistance in these patients. After daratumumab treatment, light chain normalization and complete hematologic remission were reported [84]. These reports initiated a series of studies that simultaneously confirmed the safety and tolerability of daratumumab in AL amyloidosis [82].

## 7. Conclusions

Cardiac amyloidosis is a difficult disease to diagnose and treat. AL-CA is mostly detected at an advanced stage of the disease, which is associated with the presence of organ complications at the time of diagnosis. Free light chain levels have been shown to be a prognostic indicator of survival in patients with AL amyloidosis. It is believed that not only organ amyloid deposits, but also their circulating precursors, play an important role in the damage leading to organ dysfunction. Early diagnosis and identification of the type of monoclonal chain seems to be a key issue for improving the prognosis of patients, in addition to research on new therapies. The development of various diagnostic techniques, including mass spectrometry, enables earlier diagnosis and improves patient prognosis. Despite many studies that have improved the knowledge of AL-CA treatment in addition to diagnosis, the disease remains a challenge for physicians and diagnosticians, and the prognosis of most patients remains bad.

## Figures and Tables

**Figure 1 jcm-11-00911-f001:**
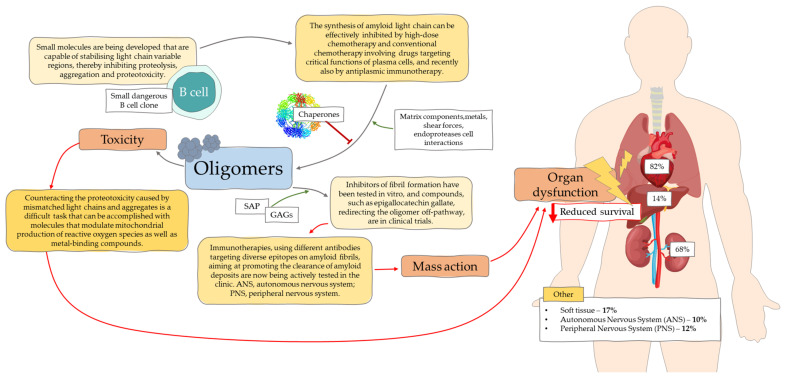
Schematic depicting the molecular mechanisms leading to light chain amyloidosis [based on [11]]. SAP, serum amyloid P; GAGs, glycosaminoglycans; ANS, autonomous nervous system; PNS, peripheral nervous system.

**Table 1 jcm-11-00911-t001:** Pathogenic proteins contributing to different types of systemic amyloidosis (based on [9]).

Name of Protein	Type of Systemic Amyloidosis
Immunoglobulin light chain	Light chain
Transthyretin (wild-type)	TRwt
Transthyretin (mutant)	TTRv
Serum amyloid A (SAA)	AA
Leucocyte chemotactic factor 2 (LECT2)	ALECT2
Gelsolin	AGel
Apolipoprotein AI (ApoAI)	AApoAI
Apolipoprotein AII (ApoAII)	AApoAII
Apolipoprotein AIV (ApoAIV)	AApoAIV
Apolipoprotein CII (ApoCII)	AApoCII
Apolipoprotein CIII (ApoCIII)	AApoCIII
Fibrinogen	AFib
β2 microglobulin	Aβ2M
Lysozyme	ALys

**Table 2 jcm-11-00911-t002:** Red flags regarding cardiac and extracardiac manifestations of AL amyloidosis (based on [65]). ECG, electrocardiogram; LV, left ventricular; AV, atrio-ventricular; NT-proBNP, N-terminal pro-B-type natriuretic peptide; HF, heart failure.

Type	Red Flag
Extracardiac	Skin bruising
	Macroglossia
	Renal insufficiency
	Proteinuria
Cardiac	Hypotension or normotensive if previous hypertensive
	Pseudoinfarct pattern in ECG
	Low/decreased QRS voltage to degree of LV thickness
	AV conduction disease
	Disproportionally elevated NT-proBNP to degree of HF
	Persisting elevated troponin levels
	Granular sparkling of myocardium in echocardiogram
	Increased right ventricular wall thickness
	Increased valve thickness
	Pericardial effusion
	Reduced longitudinal strain with apical sparing pattern
	Subendocardial late gadolinium enhancement
	Elevated native T1 values
	Increased extracellular volume
	Abnormal gadolinium kinetics

**Table 3 jcm-11-00911-t003:** Stages of cardiac involvement in AL amyloidosis [74].

Stage	Amount of Parameters	5-Year Survival
I	0 parameters	68%
II	1 parameters	60%
III	2 parameters	28%
IV	3 parameters	14%

**Table 4 jcm-11-00911-t004:** Therapeutic opportunities in AL amyloidosis.

Supportive Therapy	Therapy Specific to AL
patient education	dexamethasone with an alkylator
maintaining adequate blood pressure	cyclophosphamide with thalidomide
balancing peripheral edema and renal insufficiency	dexamethasone with thalidomide
balance between salt restriction and water supply	combinations with bortezomib
fludrocortisone enhances fluid retention	high-dose chemotherapy with autologous stem cell transplantation (ASCT)
midodrine and compression stockings for orthostatic hypotension	daratumumab

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
