# Peer review of "Physiology, Diagnosis and Treatment of Cardiac Light Chain Amyloidosis"

_jcm, 2022, doi:10.3390/jcm11040911_

Round 1

Reviewer 1 Report

Based on the abstract, the authors provide an overview on AL amyloidosis epidemiology, disease onset, diagnosis and subsequent treatment focussing on the heart. however, while reading the full MS it is far from complete.

Major concerns:

what is the purpose of the authors? a more technical review on amyloid fibril composition in relation to clinic? or merely an overview of the clinical insights in AL and AL-CA? it is farm from complete in its current state and lacks focus

  1. MGUS vs AL. Please pay attention to the difference as these are two totally different diseases. In the abstract it is stated that MGUS is resonsible for AL-CA which is absolutely not the case!
  2. I don't understand why the authors should mention table I at all
  3. Characteristics of amyloid: why not focus on AL? please also elucidate more on the pathogenesis of AL and its consequenties, please mention also pmid 31359320, pmid 34534463 (in depth on structure), overivew amyloidosis general from the London group 10.1111/joim.13169
  4. if part 3 is the main part on AL, please make this paragraph an extensive and complete overview
  5. Please pay attention in the diagnosis paragraph to the imprortance of early diagnosis pmid 31359320 including the new consensus paper in cardiac amyloidosis pmid 33825853
  6. please also add a paragrpah on prognosis, prognostification scores.
  7. if the MS focusses on AL-CA, it should also include specific cardiac diagnostics (echo, MRI, ECG). how about cardiac markers
  8. I don't see information on organ response, especially cardiac, and how to monitor patients after SCT.

Author Response

Dear Reviewer,

Thank you for your kind suggestions about our paper. We did our best to improve the manuscript by replying to all your concerns. Please find the point-by-point answers below:

Based on the abstract, the authors provide an overview on AL amyloidosis epidemiology, disease onset, diagnosis and subsequent treatment focussing on the heart. however, while reading the full MS it is far from complete.

Major concerns:

what is the purpose of the authors? a more technical review on amyloid fibril composition in relation to clinic? or merely an overview of the clinical insights in AL and AL-CA? it is farm from complete in its current state and lacks focus

  1. MGUS vs AL. Please pay attention to the difference as these are two totally different diseases. In the abstract it is stated that MGUS is resonsible for AL-CA which is absolutely not the case!

Thank you very much for pointing out this error. The fragment about MGUS in the abstract has been removed.

  1. I don't understand why the authors should mention table I at all

Our idea was to illustrate to readers what proteins may be involved in the pathogenesis of amyloidosis. Your comment prompted us to change the concept and the table was reworked. In this version of the table, we have shown the precursor proteins and the types of systemic amyloidosis in which they occur.

  1. Characteristics of amyloid: why not focus on AL? please also elucidate more on the pathogenesis of AL and its consequenties, please mention also pmid 31359320, pmid 34534463 (in depth on structure), overivew amyloidosis general from the London group 10.1111/joim.13169

Thank you very much for your comment and for presenting interesting and valuable publications. Following your advice, we have decided to focus more on AL. For this reason, the chapter "characteristics of amyloid" has been partly moved to the chapter on characterization, and the sections that we considered redundant have been removed. The advice regarding a more detailed description of the pathogenesis of AL and its consequences has been implemented under your fourth comment.

  1. if part 3 is the main part on AL, please make this paragraph an extensive and complete overview

We aimed to present AL amyloidosis as the most common type of cardiac amyloidosis. An important element for us was to present that AL amyloidosis does not affect only the heart, but is a multi-organ disease. We hope we have achieved this by creating a comprehensive chapter divided into smaller paragraphs covering separately the symptoms, mechanism of onset, and general description of the disease. But to make it more apparent that this is the main part of our work we have included AL cardiac amyloidosis as part of Chapter 3 and added sections to emphasize the relevance of the topic we are working on.

  1. Please pay attention in the diagnosis paragraph to the imprortance of early diagnosis pmid 31359320 including the new consensus paper in cardiac amyloidosis pmid 33825853

At the beginning of the chapter, we emphasized the importance of early diagnosis of the disease.

  1. please also add a paragrpah on prognosis, prognostification scores.

We have added as a fifth chapter additional information about patient prognosis and have included a section we have already written about the Mayo Clinic classification.

  1. if the MS focusses on AL-CA, it should also include specific cardiac diagnostics (echo, MRI, ECG). how about cardiac markers

We have expanded the chapter on diagnostics by including an additional chapter focusing on cardiac imaging.

  1. I don't see information on organ response, especially cardiac, and how to monitor patients after SCT.

We have added a section on SCT along with a description of the effectiveness of this therapy and how patients are classified for it.

Once again, we would like to thank you for the contribution, and we do hope that our manuscript can be accepted in the current form.

Kind regards,

Paulina Niedźwiedzka-Rystwej

Reviewer 2 Report

I read the manuscript entitled "Physiology, diagnosis and treatment of cardiac light chain amyloidosis ". Here the author provided a nice and comprehensive overview of the topic. I have the following comments:

  • Diagnosis paragraph: I would include some more sentences on cardiovascular imaging findings in AL cardiac amyloidosis (eg cardiac magnetic resonance tissue characterization/mapping, echocardiographic features etc.). A table summarizing the main cardiac diagnostic findings would enhance readability (eg ECG, echo, CMR etc.) 
  • - Therapy- this paragraph would benefit from a Table too for summarizing contents

- few typos (eg line 73)

Author Response

Dear Reviewer,

Thank you for your kind suggestions about our paper. We did our best to improve the manuscript by replying to all your concerns. Please find the point-by-point answers below:

I read the manuscript entitled "Physiology, diagnosis and treatment of cardiac light chain amyloidosis ". Here the author provided a nice and comprehensive overview of the topic. I have the following comments:

  • Diagnosis paragraph: I would include some more sentences on cardiovascular imaging findings in AL cardiac amyloidosis (eg cardiac magnetic resonance tissue characterization/mapping, echocardiographic features etc.). A table summarizing the main cardiac diagnostic findings would enhance readability (eg ECG, echo, CMR etc.) 

We have expanded the chapter on diagnostics by including an additional chapter focusing on cardiovascular imaging. We have also included a summary table about the red flags in the diagnosis of AL amyloidosis

  • - Therapy- this paragraph would benefit from a Table too for summarizing contents

We have also included a table at the end of this chapter summarizing the therapeutic options in cardiac AL amyloidosis

- few typos (eg line 73)

All typos we found have been corrected.

Once again, we would like to thank you for the contribution, and we do hope that our manuscript can be accepted in the current form.

Kind regards,

Paulina Niedźwiedzka-Rystwej

Round 2

Reviewer 1 Report

No further comments, the ms revisions are adequate although the MS still remains rather limited withouth addressing the latest up-to-date therapy like daratumumab

Author Response

Dear Reviewer,

Thank you for your kind consideration. In the current form of the manuscript we have we have added a final chapter on the latest treatment with daratumumab for patients diagnosed with AL amyloidosis who have demonstrated resistance to other treatments.

We hope this will satisfy you.

Kind regards,

Paulina Niedźwiedzka-Rystwej